# Identification of Emerging Hazards in Mussels by the Galician Emerging Food Safety Risks Network (RISEGAL). A First Approach

**DOI:** 10.3390/foods9111641

**Published:** 2020-11-10

**Authors:** Marta López Cabo, Jesús L. Romalde, Jesus Simal-Gandara, Ana Gago Martínez, Jorge Giráldez Fernández, Marta Bernárdez Costas, Santiago Pascual del Hierro, Ánxela Pousa Ortega, Célia M. Manaia, Joana Abreu Silva, Juan Rodríguez Herrera

**Affiliations:** 1Seafood Microbiology and Technology Section, Instituto de Investigacións Mariñas, Spanish National Research Council (CSIC), 36208 Vigo, Spain; mbernar@iim.csic.es (M.B.C.); spascual@iim.csic.es (S.P.d.H.); juanherrera@iim.csic.es (J.R.H.); 2Department of Microbiology and Parasitology, CIBUS-Faculty of Biology & Institute CRETUS, Universidade de Santiago de Compostela, E15782 Santiago de Compostela, Spain; jesus.romalde@usc.es; 3Nutrition and Bromatology Group, Department of Analytical and Food Chemistry, Faculty of Food Science and Technology, Universidade de Vigo–Ourense Campus, E32004 Ourense, Spain; jsimal@uvigo.es; 4Department Analytical and Food Chemistry, Universidade de Vigo, 36310 Vigo, Spain; anagago@uvigo.es (A.G.M.); jgiraldez@uvigo.es (J.G.F.); 5Direccion Xeral de Innovación e Xestión da Saúde Pública, Consellería de Sanidade, Xunta de Galicia, 15781 Santiago de Compostela, Spain; anxela.pousa.ortega@sergas.es; 6CBQF—Centro de Biotecnologia e Química Fina—Laboratório Associado, Universidade Católica Portuguesa, Escola Superior de Biotecnologia, Rua Diogo Botelho 1327, 4169-005 Porto, Portugal; cmanaia@porto.ucp.pt (C.M.M.); jsilva@porto.ucp.pt (J.A.S.)

**Keywords:** food safety, risks, bivalves

## Abstract

Emerging risk identification is a priority for the European Food Safety Authority (EFSA). The goal of the Galician Emerging Food Safety Risks Network (RISEGAL) is the identification of emerging risks in foods produced and commercialized in Galicia (northwest Spain) in order to propose prevention plans and mitigation strategies. In this work, RISEGAL applied a systematic approach for the identification of emerging food safety risks potentially affecting bivalve shellfish. First, a comprehensive review of scientific databases was carried out to identify hazards most quoted as emerging in bivalves in the period 2016–2018. Then, identified hazards were semiquantitatively assessed by a panel of food safety experts, who scored them accordingly with the five evaluation criteria proposed by EFSA: novelty, soundness, imminence, scale, and severity. Scores determined that perfluorinated compounds, antimicrobial resistance, *Vibrio parahaemolyticus*, hepatitis E virus (HEV), and antimicrobial residues are the emerging hazards that are considered most imminent and severe and that could cause safety problems of the highest scale in the bivalve value chain by the majority of the experts consulted (75%). Finally, in a preliminary way, an exploratory study carried out in the Galician Rías highlighted the presence of HEV in mussels cultivated in class B production areas.

## 1. Introduction

Identification of risks at early appearance (emerging) is a matter of public health, as it can be a major preventive instrument for foodborne diseases [1]. That is why the European (EU) EU Food Law (Art. 34 Reg. 178/2002) established the emerging risk identification (ERI) in food and feed as a requirement for the authorities. In this framework, the European Food Safety Authority (EFSA) coordinates an Emerging Risk Exchange Network (EREN) with the mission of identifying the main emerging issues in food and feed safety in Europe. In a networking way, all national emerging risks networks of the member states provide information to EREN of any signal of emerging hazards in their respective countries.

The Galician Food Safety Emerging Risks Network (RISEGAL) was born in 2018 with the mission of contributing to the identification of food safety emerging hazards and their driving factors in foods produced or commercialized in Galicia.

Galicia is a small region located in the northwest corner of the Iberian Peninsula. The food industry is one of the main economic sectors of the region. In fact, Galicia leads in seafood production and processing in the EU, and local industries constitute half of the national industry of the sector [2]. In particular, the mussel industry is the largest productive activity of Galicia, with an estimated production of 250,000 tons of mussels per year, of which 225,000 tons are marketed. Beside mussels, oysters, clams, and cockles are also important primary products of the region. This high production has consolidated a highly profitable long-term stable market for Galicia [3].

Bivalve mollusks are recognized as vectors in the transmission of viral and bacterial diseases [4]. Therefore, a continuous sanitary control, at both production and retail levels, as well as several regulation actions, is established in order to minimize the risks of disease transmission [5].

Emerging risks in food safety are promoted by socioeconomic and environmental driving factors, such as consumption habits, migration, increasing ageing, and climate change. Climate change is a recognized challenge to food safety. Changes in temperature, humidity, rainfall patterns, and increases in weather events affect all productive systems. These impacts will affect the persistence and occurrence of germs and toxin production microorganisms (like algae and fungi) and the patterns of foodborne diseases, thus increasing the risk of emerging zoonosis. Additionally, indirect effects derived from the application of mitigation strategies are expected [6].

Most driving factors can be considered nonglobal but dependent on each specific geographical location. In Galicia, for example, ageing is considered an important driving factor for the appearance of emerging hazards. Also, traditional habits of shellfish consumption, raw or undercooked, undoubtedly increase foodborne illness risks [7]. Most bivalves are grown in floating rafts located in fiordlike inlets near the coast, called rías, where there is high nutrient concentration and often sewage contamination. These specific conditions should be considered in the emerging hazards identification procedure.

Consequently, the aim of this work was to identify and prioritize emerging hazards potentially associated with bivalve shellfish produced in Galicia. Additionally, an exploratory study of such hazards was subsequently performed to find out whether they could be already detected in mussels cultured in Galician Rías from harvesting areas approved for human consumption.

## 2. Materials and Methods

### 2.1. Online Survey

A short (six questions) online inquiry was designed by using the free software Typeform (platform provided by Typeform SL, (the platform accessible though the www.typeform.com domain name (the “site”) is provided by TYPEFORM SL, Barcelona, Spain. The inquiry was addressed to public health inspectors, consumers, and stakeholders of the food industry, and responses were automatically collected by the program in an optional anonymous manner.

### 2.2. Nonscientific Survey

A nonscientific literature survey was performed by using FoodRiskScan, a technological solution developed by a Spanish food technological center called AINIA that can automatically scan “groups of concepts” in different “searching groups” in nonscientific online sources previously defined (Appendix A). Outputs were compiled and analyzed by the experts of RISEGAL.

### 2.3. Scientific Survey

Members of RISEGAL performed a scientific survey through the Web of Science (WOS) Core Collection database. Two consecutive searching steps (general and specific) were carried out by using the keywords included in Appendix A. Articles obtained from the general search boots criteria used to carry out the specific search. Outputs were further submitted to prioritization and assessment steps by the panel of experts of RISEGAL.

To gather data from the different stakeholders concerned with the mollusk bivalve chain, data collection was carried out by these three complementary approaches.

### 2.4. Preliminary Assessment by Scoring

Members of RISEGAL and 10–15 additional external experts carried out a blind scoring (0–5) of novelty, severity, imminence, soundness, and scale, all criteria considered by EFSA for the initial qualitative description of emerging issues [8].

### 2.5. Exploratory Study

#### 2.5.1. Shellfish Sampling

In each sampling time, mussels were obtained from three sites located in different zones in Galician Rías (Ría de Arousa, Ría de Pontevedra, and Ría de Vigo) (Figure 1).

They were collected monthly from May to July 2019. The sampling points (Table 1) were located in class B production areas (230–4600 MPN *Escherichia coli* per 100 g shellfish) according to European legislation. Mussels were transported alive on ice to the laboratory and distributed in groups of 15–20 individuals for further analysis.

#### 2.5.2. Determination of *Vibrio parahaemolyticus* and *Vibrio vulnificus*

*Vibrio parahaemolyticus* and *Vibrio vulnificus* were detected following the ISO 21872-1:2017 method. Mussels were washed, scrubbed, and opened aseptically. Flesh and intravalvar fluids were collected and homogenized for 120 s in a sterile bag using a Stomacher (IUL, S.A., Barcelona, Spain). Enrichment was carried out in two subsequent steps. First, 25 g of the homogenized sample was homogenized in 225 mL of alkaline saline peptone water (ASPW) and incubated at 37 °C for 6 h. Second, 1 mL was reinoculated into 10 mL of ASPW and incubated at 37 and 45 °C for 18 h. At the end of the incubation period, an inoculating loop of the culture broth was streaked onto thiosulfate citrate bile salt sucrose (TCBS, ISO Scharlau, Spain) and Vibrio ChromoSelect agar (Sigma-Aldrich, St. Louis, MO, USA). After incubation at 37 °C for 24 h, five typical colonies were subcultured on saline nutrient agar (3% NaCl) plates. Presumptive isolates were confirmed as *V. parahaemolyticus* by PCR amplification of the *toxR* and enteropathogenicity-associated genes (*tdh* and *trh*) or the *vvh* gene in the case of *V. vulnificus*.

#### 2.5.3. Determination of Cultivable Heterotrophic Bacteria

Twenty-five grams of bivalve flesh obtained from 15–20 mussels were homogenized in 225 mL of 0.1% peptone water (Scharlab, Sentmenat, Spain) for 60 s. Serial dilutions were streaked on marine agar plates (Scharlab, Sentmenat, Spain) and incubated at 25 °C for 5 days [9]. Results were expressed in colony-forming units of culturable heterotrophic bacteria per gram (CFU/g).

#### 2.5.4. Isolation of Antibiotic-Resistant Bacteria

Twenty-five grams of bivalve flesh obtained from 15–20 mussels was homogenized for 2–3 min in a Stomacher (IUL, S.A., Barcelona, Spain) in 100 mL of TSB (Scharlau, Spain) supplemented with cicloproxacin (4 mg/L) and 1% sodium chloride. This fluoroquinolone was selected because of the widespread distribution of resistance against this antibiotic [10]. Enrichment of the culture was carried out by incubation at 25 °C for 20 h. In a second phase, a volume of 200 µL of the enriched culture was streaked on modified Trytone Soy Agar (TSA) plates and incubated for 72 h at 30 °C. After incubation, individual colonies were purified, identified, and screened for the presence of antibiotic resistance genes.

#### 2.5.5. Identification of Ciprofloxacin-Resistant Bacteria

Identification of colonies with low susceptibility to ciprofloxacin was based on 16S rRNA gene sequence and analysis following the procedure described by Rodríguez-López et al. [11].

#### 2.5.6. Detection of Antibiotic Resistance Genes

Antibiotic resistance gene detection was carried out by conventional PCR as described before [10]. The set of genes that were screened was selected based on their wide environmental distribution and because they represented distinct antibiotic classes *–bla_TEM_*, *bla_CTX-M_* (beta-lactams), *qnrS* (quinolones), *sul1* (sulphonamides), and *aac6* (aminoglycosides). In addition, we screened the gene *intI1*, which encodes for an integrase enzyme associated with class 1 integrons, recognized as a proxy of anthropogenic microbial sources [12]. 

#### 2.5.7. Determination of Hepatitis E Virus HEV in Mussels

Viral recovery from shellfish homogenates (2 g) was performed according to ISO 15216-1:2017, with slight modifications [13]. Briefly, known amounts (10 µL, 10^3^ PFU) of Mengovirus (clone vMC0) were spiked into each homogenate as RNA extraction efficiency control. One volume of 0.1% peptone water pH 7.5 (1:1 *w*/*v*) was added to each homogenate, which was then shaken for 1 h at 4 °C and centrifuged at 1000× *g* for 5 min, recovering the supernatant. Viral RNA was extracted from the supernatants (150 μL sample volume) using the NucleoSpin^®^ RNA Virus Kit (Macherey-Nagel, Duren, Germany) in duplicate, following the manufacturer’s instructions.

RT-qPCR targeting the ORF3 region of HEV [14] was employed for the amplification of viral RNA extracts (undiluted and diluted 1/10). Appropriate negative (containing no nucleic acid) and positive (viral RNA) controls were included in each run. Amplification was performed using the iTaq Universal Probes One-Step Kit (Bio-Rad, Hercules, CA, USA) with primers JVHEVF/JVHEVR and probe JVHEVP, a TaqMan^®^ probe containing a 5′ 6-carboxy fluorescein fluorophore and 3′ Black Hole Quencher. Reverse transcription was carried out at 50 °C for 15 min, followed by denaturation at 95 °C for 5 min. cDNA was then amplified with 45 PCR cycles at 95 °C (15 s) and 55 °C (20 s). Extraction and amplification efficiencies were calculated using Mengovirus and appropriate external controls (quantified HEV RNA from a clinical sample) as indicated in the ISO 15216-1:2017 protocol.

Viral genome copies were quantified accordingly to the ISO 15216-1:2017. Briefly, serial dilutions of HEV RNA purified from a clinical sample (kindly donated by Dr. A. Aguilera from the University Hospital of Santiago de Compostela, Spain) were employed to construct standard curves, plotting the number of genome copies against the Cq values. Results were expressed as number of RNA viral genome copies per gram of digestive tissue.

#### 2.5.8. Determination of Antibiotic Residues in Mussels

It was carried out following the method of Chiesa et al. [15]. Mussels were thawed and allowed to drain for 24 h to remove as much water as possible. Once drained, the entire sample was homogenized with a chopper or similar. One gram of homogenized sample was taken in a 15 mL polypropylene tube in duplicate, and 100 µL of a 20% trichloroacetic acid solution and 5 mL of Ethylenediaminetetraacetic acid EDTA–McIlvaine buffer solution at pH 4 (Na_2_HPO_4_·2H_2_O, citric acid, and EDTA) were added, mixed in vortex, and placed in an ultrasonic bath for 20 min. Samples were centrifuged (7500 rpm, 4 °C) for 10 min, and supernatant was transferred to another polypropylene tube. After adding 3 mL of hexane, the mixture was stirred and centrifuged for 5 min. Following centrifugation at 2500× *g* and 4 °C for 5 min, supernatant was taken to a PTFE tube for defatting with hexane twice. Following centrifugation as before, hexane was removed, and the extract was cleaned up by Oasis HLB solid-phase extraction under smooth vacuum. An 85% phosphoric acid solution (2% of total volume) was added to the resulting extract, which was eluted through a solid-phase extraction (SPE) cartridge (Phenomenex Strata-X 200 mg/6 mL, Alcobendas, Spain), previously conditioned with 3 × 5 mL of methanol and 2 × 5 mL of water. After elution of the sample, the cartridge was washed with 3 × 5 mL of 5% methanol in water and dried for 1 min under vacuum. Elution was carried out with 6 mL of methanol with 2% formic acid in a 15 mL polypropylene tube. Finally, sample was concentrated to almost dryness in a vacuum evaporation system (Büchi Multivapor P-12, Essen, Germany). The resulting residue was redissolved in 1 mL of water with 0.1% formic acid in an analytical vial.

The analysis of the presence of the different antibiotics was performed by high-efficiency chromatography (Agilent 1260 Infinity, Agilent Technologies, CA, USA) coupled with a triple quadrupole mass spectrometer (AB Sciex 3500, Alcobendas, Spain). The injection volume was set at 10 μL, and the injector and column were maintained at 10 and 40 °C, respectively. Chromatographic separation was carried out on a Phenomenex Kinetex Biphenyl 1.7 μm column (50 × 2.1 mm), using mobile phase wate with 0.1% formic acid (A) and methanol with 0.1% formic acid (B). Elution was performed at a flux of 300 µL/min through the following gradient: from 5% B to 100% B in 10 min, and 2 min at 100% B, and then 0.2 min to recover initial conditions and keep it stable for 5.8 min. The mass spectrometer used an electrospray source (ESI), working in positive or negative mode depending on the compound to be determined. The ESI parameters were as follows: curtain gas, 30 L/min; collision gas, 8 L/min; ion source gas, 50 L/min; temperature, 400 °C; and ion spray voltage, 5.5 kV for ESI+ and −4.2 kV for ESI. Detection in the mass spectrometer was performed in MRM (multiple reaction monitoring) mode considering two transitions for each analyte (conditions used for its antibiotic are showed in Appendix A).

In the chromatograms obtained for the two transitions of each antibiotic, peaks with the same retention time were sought, whereas those that were not were discarded. Once the elution order of the antibiotics was known, the time range where the antibiotic in question was eluted was known. Those peaks with an identified signal-to-noise ratio (S/N) higher than 10 in the two chromatograms of the transitions would be reported as present.

#### 2.5.9. Determination of Tetrodotoxin

##### N2a Assay

N2a cell assay is based on toxicity associated with the VGSCs (voltage-gated sodium channels) resulting from the use of veratridine and ouabain. The assay enables the semiquantitation of tetrodotoxin TTX based on the percentage of living cells remaining.

The conditions used in this assay were proposed by Manger et al. [16,17] with slight modifications to accommodate the assay for the detection of TTX [18].

The same conditions described by Turner et al. [19], with slight modifications [20], were used for the extraction of TTX for both N2a and LC–MS/MS, and the last step of dilution with acetonitrile required for the LC–MS/MS analysis was not carried out for N2a to avoid possible interferences in the cell assay.

Cell viability was measured by a colorimetric method using tetrazolium (MTT) metabolism [18].

##### LC–MS/MS Analysis

###### Extraction and SPE-ENVI-Carb Clean-Up

The extraction method conditions described by Turner et al. [19] and the clean-up conditions described by Boundy et al. [21], both with slight modifications included in EURLMB TTX SOP [20], were used in this work.

###### HILIC LC–MS/MS Analysis

An Agilent 1290 Infinity LC system (Agilent Technologies Deutschland GmbH, Waldbronn, Germany) was used for the liquid chromatographic separation. This separation is described in Leão et al. [18]. The chromatographic conditions used in the analysis of TTX are summarized in Appendix A.

A 6460A Triple Quadrupole mass/massMS/MS (QQQ) equipped with a Jet Stream ESI source (Agilent Technologies Deutschland GmbH Waldbronn, Germany) was used for the analysis of TTX in MRM (multiple reaction monitoring) mode by detecting m/z transitions in tandem mass spectrometry, which are included in Appendix A [20].

The optimized conditions are summarized in Appendix A [18].

## 3. Results and Discussion

### 3.1. Identification of Food Safety Emerging Problems in Bivalve by RISEGAL

The procedure followed by RISEGAL for the identification of emerging hazards in bivalves is schematized in Figure 2. It comprises three steps: data collection, prioritization of emerging hazards, and preliminary evaluation of emerging hazards. It was adopted from the official emerging risks identification (ERI) procedure initially defined by EFSA [1] and further modified by including an initial key step consisting in the identification of emerging issues through EFSA networks (EREN, StaDG-ER, scientific panels, etc.) [22].

### 3.2. Data Collection

#### 3.2.1. Online Survey

A short online inquiry (six questions) was designed in order to gather information related to which emerging issues/problems could be identified by different actors associated with the production and marketing of bivalves in Galicia. RISEGAL invited professionals of the industrial sector as well as health professionals, veterinary inspectors, and consumers. Questions and answer options are included in Table 2.

Metrics data are included in Appendix A. Only 41% of persons who initiated the inquiry were able to conclude it. In fact, whereas the software registered 214 visitors to the inquiry, only 50 responses (23%) were achieved, which is probably due to the difficulties found in responding to Q2, which requires an explicit and descriptive answer about a supposed identified emerging hazard. 

Veterinary inspectors, researchers, and professionals of health surveillance answered theaccounted for 56% of the responses. Workers associated with the sector of food industry provided 20% of the responses, being the rest of responses answered by citizensand the rest of the responses were completed by citizens. From those 50 different answers collected since august 2018, 35 reported possible emerging food safety problems in bivalve mollusks (70%; Q1), and 54% of them were associated with mussels (Q3). Some of the most frequently reported emergent problems were related to the presence of HEV in mussels, norovirus in oysters, increase of *Vibrio* spp. in clams or mussels, presence of persistent organic contaminants (POPs) and heavy metals in mussels, and increase of problems associated with red tides. Most of the respondents (~30%) identified virus as the principal causative agent, followed by bacteria, chemical contaminants, and biotoxins.

#### 3.2.2. Nonscientific Literature Survey

A nonscientific literature survey was performed by using FoodRiskScan as described in Materials and Methods (Section 2.2). The scanning robot was run according to the concepts, sources, and searching groups identified. The automatic search rendered 591 outputs that were subsequently filtered by the experts of RISEGAL. As a result, 165 outputs were finally selected as closely related to the topics of interest. However, most of the outputs were associated with safety problems in oysters in the USA (“oyster” was named in more than 100 outputs out of 165; see Appendix A), thus indicating oyster-related publications distorted the final results obtained.

#### 3.2.3. Scientific Literature Survey

The general and specific scientific searches rendered a total of 251 and 751 articles, respectively. After several subsequent filtering steps, 126 articles were selected by the experts as directly related to emerging risks associated with bivalves. Those 126 articles were grouped by RISEGAL per type of hazard for further evaluation: biotoxins (17), parasites (14), virus (15), chemical contaminants (28), bacteria (38), and antimicrobials (14). 

A comparison between the outputs rendered by the three search strategies used for data collection demonstrated that information obtained through the scientific survey already included those obtained from both the online inquiry and the nonscientific survey. Consequently, only those outputs from the scientific survey were considered in further steps. This was in agreement with EFSA panels that have already pointed out the lack of efficiency of the nonscientific (grey) literature for the identification of signals related to emerging risks [8].

### 3.3. Prioritization of Emerging Hazards

The scientific groups of articles selected were sent to the experts of RISEGAL for them to select a maximum of 3 emerging hazards according to a brief form previously designed (Appendix A). Additional information, such as type of emerging hazards, cause of the appearance, associated driving factors, and probability of incidence in Galicia, was also requested. Results are shown in Table 3. As can be observed, a total of 11 emerging hazards were prioritized by the experts, the majority of biological origin (8 out of 11). Changes in consumption habits and environmental factors were the driving factors more frequently identified by the experts.

### 3.4. Preliminary Evaluation of Emerging Hazards

#### 3.4.1. Scoring

Outputs were gathered, grouped in two intervals (0–2 and 3–5), and analyzed by using the relative frequency of responses (Figure 3). By consensus, RISEGAL defined that those emerging issues scored by 75% of the experts in the interval 3–5 with respect to imminence, scale, and severity were most relevant. As a result, it was concluded that perfluorinated compounds, antimicrobial resistance, *Vibrio parahaemolyticus*, HEV, and antimicrobial residues are the emerging hazards that are considered most imminent and severe and that could cause safety problems of the highest scale in the bivalve value chain.

Perfluorinated compounds (PFCs) are toxic persistent environmental pollutants. They are mainly used as flame retardants in many commercial products and, for this reason, are easily found in aquatic life [23] and water reservoirs [24], ranging from 0.06 to 10.9 ng/L in water, 0.01–0.13 ng/g dw in sediments, and 0.01–0.06 ng/g ww in mussels.

Antimicrobial resistance (AMR) is one of the greatest challenges of the 21st century [25]. According with the World Health Organization (WHO), addressing AMR requires a One Health holistic approach involving humans, animals, and environment since resistant bacteria may spread without border restrictions. Antibiotic-resistant bacteria, once disseminated in the environment, can survive and even proliferate, with the capability of colonizing new habitats, namely, the food chain [26,27]. Therefore, it can be hypothesized that mussels grown in Galician Rías can be exposed to and contaminated with bacterial fecal pathogens. In fact, mussels are successfully used as indicator organisms in marine pollution monitoring. Besides the effects on the development of antibiotic-resistant bacteria, antibiotics could have toxic effects on aquatic animals and, at low concentrations, could act as signaling agents and even change the natural microbial diversity in aquatic ecosystems [24].

*V. parahaemolyticus* is a typical warm-water bacterial pathogen and a worldwide leading cause of bacterial illness associated with seafood consumption [28]. Although it is not considered a frequent pathogen in Europe, the exception is Galicia, where several outbreaks of *V. parahaemolyticus* have been described since 1990 [18]. Moreover, a long-term previous study evidenced that climate change is influencing the occurrence of *Vibrio* infections in the world [29]. The importance of *V. parahaemolyticus* as an emerging hazard becomes even higher when considering that some authors attribute the biosynthesis of TTX to *Vibrio* [30].

HEV belongs to the family Hepeviridae within the genus *Orthopevirus*, which includes five genotypes that infect humans (HEV 1, 2, 3, 4, and 7) [31]. Of these, genotypes 1 and 2 are prevalent in developing countries in Asia, Africa, and Central America. These genotypes are mainly restricted to humans and are transmitted by consumption of fecal-contaminated water in areas with poor sanitation [32]. On the other hand, genotypes 3 and 4, found in industrialized countries, are confirmed as the major cause of zoonotic HEV, with pigs and wild boards as the main reservoirs [31]. HEV is known to bypass wastewater treatment plants, and as consequence, coastal waters can be contaminated with HEV of swine and human origin being further concentrated by bivalve mollusks during filtration and giving rise to a fecal–oral transmission route [33].

#### 3.4.2. Exploratory Study

The presence of antibiotic residues, HEV, *V. parahaemolyticus*, *V. vulnificus*, antibiotic-resistant bacteria (ARB), and TTX was determined in mussels collected at class B production areas in the Rías of Galicia. Obtained results demonstrated the absence of *V. parahaemolyticus*, *V. vulnificus*, and TTX in the mussels collected in the different zones of production. No antibiotic residues were detected in mussel samples. HEV and some ciprofloxacin-resistant bacteria were detected.

Hepatitis E virus

The HEV genome was detected in mussels from Vilagarcía A and Pobra E production zones in May and June, with a considerably higher number of copies found in May (around 10^3^ copies/g of digestive tissue versus 15 copies/g in June) in both zones. HEV was not detected in samples collected in July (Table 4).

In territories with a high density of farming, besides contamination with human waste, coastal waters can be contaminated since runoff following manure application can occur [34,35]. Shellfish bioaccumulates environmentally enteric viruses from contaminated coastal waters, and its role as vehicle for these agents has been well established, namely, for hepatitis A virus and noroviruses [33,36]. For HEV, the role of shellfish as vehicle of transmission has recently been considered by the scientific community. Thus, HEV has been detected in shellfish in many European countries like France [33], Denmark [37], Italy [38,39,40], Spain [13,35], and Scotland [41]. In addition, some studies have linked the consumption of shellfish to hepatitis E infections in Vietnam and Japan [42,43], and also on a cruise ship [44].

Although further studies are needed to clearly determine the public health significance of HEV detection in shellfish, the evidence obtained here reinforces the need for the inclusion of this virus in risk assessment protocols for bivalve mollusks.

Antibiotic-resistant bacteria (ARB)

The abundance of culturable heterotrophic bacteria (CHB) and ciprofloxacin-resistant bacteria (CRB) obtained in mussel samples collected at different B production zones are shown in Table 5. Abundance of CHB in the different months sampled ranged between 5.2 log CFU/g obtained in mussels collected in June and 6.5 log CFU/g obtained in May. After a 24 h sample enrichment in the presence of ciprofloxacin, the number of CFU varied between <15 and 500 colonies/g of mussel in different samples.

These results are in agreement with previous works, in which *A. rivipollensis* was first isolated from biofilm and sediment samples of the Ter River in the framework of an antibiotic-resistant bacteria study [45] and further described as a new species of *Aeromonas autochthonous* in aquatic environments [46]. Also, *A. rivipollensis* and *E. coli* revealed the presence of the gene *intI1*, which has been considered a good proxy for pollution [12]. It should be taken into account that this is a very preliminary study that needs future research to settle any conclusion regarding the presence of antibiotic-resistant bacteria in mussels collected in Galicia.

## 4. Conclusions

Emerging hazards in food safety should be foreseen to implement effective mitigation strategies on time. It is the only way to be prepared against oncoming changes, such as those associated with climate change. This work constitutes an example of an initial approach to emerging hazard identification in the bivalve shellfish value chain, which was selected for its socioeconomic impact in Galicia. Although further exploratory studies will be carried out in future research, the present results permitted the study to conclude that perfluorinated compounds, antimicrobial resistance, *Vibrio parahaemolyticus*, and HEV antimicrobial residues are the emerging hazards that are considered most imminent and severe and that could cause safety problems of the highest scale. An exploratory phase showed the presence of hepatitis E in mussels collected in different production zones of Galician Rías, whereas TTX, *V. parahaemolyticus, V. vulnificus*, and antibiotic residues were not detected. Future research should be directed towards the risk assessment of identified hazards too. RISEGAL’s mission will proceed to focus also on other value food chains in order to identify new emerging hazards that could compromise food safety in our region.

## Figures and Tables

**Figure 1 foods-09-01641-f001:**
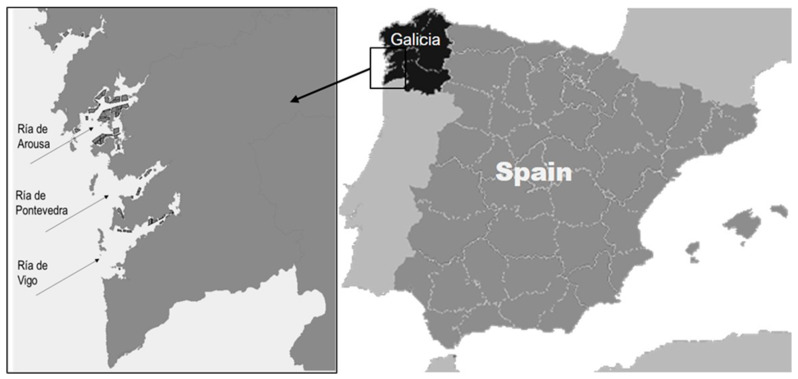
Geographic location of the sampling points considered in the exploratory study.

**Figure 2 foods-09-01641-f002:**
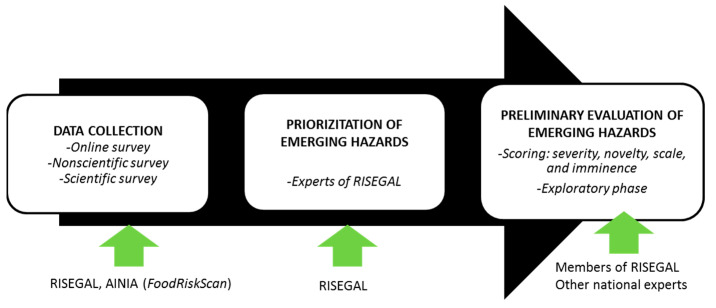
Scheme of the approach used by RISEGAL for the identification of emerging hazards.

**Figure 3 foods-09-01641-f003:**
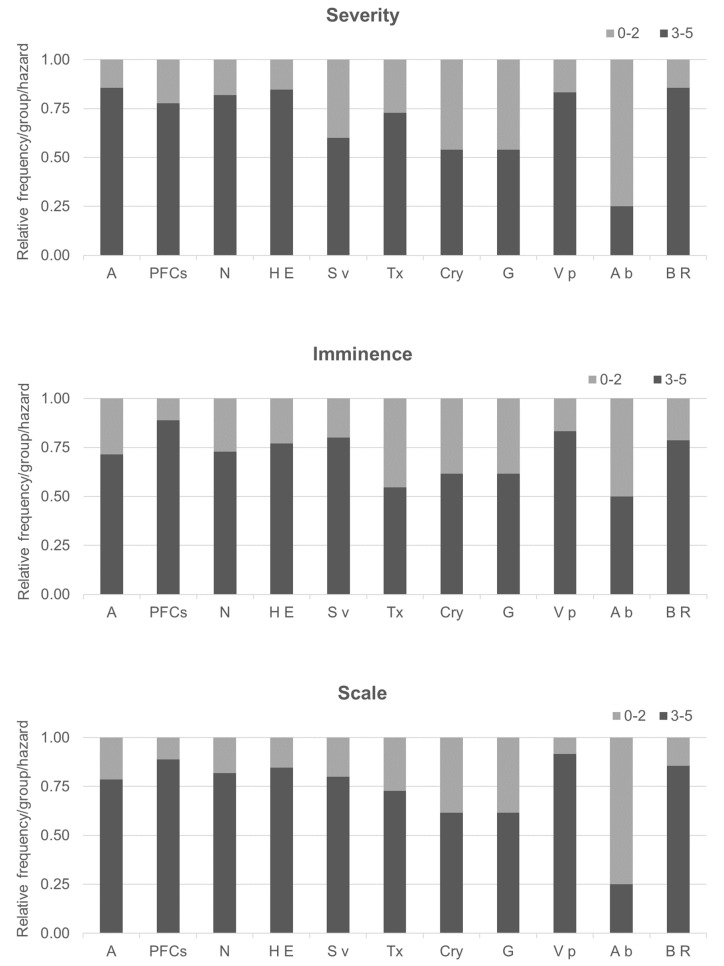
Results of the preliminary evaluation of emerging hazards by experts. A: Antimicrobials, PFCs: Perfluorinated compounds, N: Nanoparticles, HE: hepatitis E, Sv: sapovirus, Tx: tetrodotoxin, Cry: *Cryptosporidium*, G: *Giardia*, Vp: *Vibrio parahaemolyticus*, Ab: *Arcobacter butzleri*, BR: bacterial resistance.

**Table 1 foods-09-01641-t001:** Number of sampling points (*) considered in the exploratory study.

Month	Site 1	Site 2	Site 3
	Ría de Arousa	Ría de Vigo	Ría de Pontevedra
May	**		
June	**	*	*
July	**	*	*

** two sample points.

**Table 2 foods-09-01641-t002:** Questions in the online inquiry formulated by RISEGAL.

Q.1. Could you identify any “emerging problem” in food safety that could affect the production and commercialization chain of bivalve mollusks? Yes/no.
Q.2. Brief description of the identified emerging problem. Note: the answer should describe briefly the emerging problem about which she/he is thinking.
Q.3. In your opinion, which group of bivalve mollusks could be mostly affected by the identified emerging problem? Clam/mussel/oyster/other.
Q.4. What is the time scale for the identified emerging problem to occur? short term (<2 years), medium term (2–10 years), and long term (>10 years).
Q.5. Could you identify the step of the value chain most affected by the problem? Production/transformation/distribution.
Q.6. To which sector do you belong? Food industry worker/sanitary inspector or researcher/citizen.

**Table 3 foods-09-01641-t003:** List of prioritized emerging hazards by the experts of RISEGAL.

HAZARD	AGENT	NEW	CLASSIFICATION	RELATED CAUSE	VULNERABLE GROUPS	TEMPORAL SCALE	PRESENCE IN GALICIA
Drug residues in bivalvos	Antimicrobials	Yes	Increased susceptibility	Changes in consumption habits	Immunocompromised	1–3 years	Likely
Organic pollutant residues in bivalves	Perfluorinated compounds	Yes	Increased exposure	Direct human intervention	Childhood	3–10 years	Likely
Nanoparticles residues in bivalves. Transport capacity of other pollutants	Nanoparticles	Yes	Increased exposure	Changes in consumption habits	Childhood	1–3 years	Likely
Hepatitis E virus in bivalves	Hepatitis E virus	Yes	New scenarios	Control not required Scarce data	Any group	1–3 years	Likely
Sapovirus in Galician and import bivalves	*Sapovirus*	Yes	Increased exposure	Control not required Scarce data	Any group	1–3 years	Likely
Tetrodotoxin in European coast	Tetrodotoxin	Yes	New scenarios	Environmental factors	Any group	3–10 years	Insufficient data
*Cryptosporidium* in bivalves	*Cryptosporidium*	No	Incresased exposure	Control not required Scarce data	Any group	*	Insufficient data
*Giardia* in bivalves	*Giardia*	No	Undefined	Control not required Scarce data	Any group	*	Insufficient data
*Vibrio parahaemolyticus* in bivalves	*Vibrio parahaemolyticus*	No	New scenarios	Environmental factors	Any group	1–3 years	Likely
*Arcobacter* spp. in bivalves molluscs cultivated in Galicia	*Arcobacter* spp. *(A. butzuli)*	No	Increased exposure	Environmental factors	Any group	3–10 years	Insufficient data
Bivalves as reservoirs of antibiotic resistant bacteria and their relationship to the transmission of resistance genes	Antibiotic-resistant bacteria of last resort	No	Increased susceptibility	Direct human intervention	Immunocompromised and old adult	>10 years	Insufficient data

*: Not estimated by the experts.

**Table 4 foods-09-01641-t004:** Detection and quantification (in RNA copies/g) of HEV during the exploratory study.

Production Zone	May	June	July
Site 1	1.09 × 10^3^	15	n.d.
Site 2	3.01 × 10^3^	260	n.d.

n.d.: not detected.

**Table 5 foods-09-01641-t005:** Culturable heterotrophic bacteria (CHB; log CFU/g) and ciprofloxacin-resistant bacteria (CRB; CFU/g) present in mussels from different production zones of Galicia.

ORIGIN	Site 1		Site 2		Site 3	
DATE	CHB	CRB	CHB	CRB	CHB	CRB
May 2019	6.80 (0.33) 6.25 (0.22)	* *	*	*	*	*
June 2019	4.79 (0.07) 5.68 (0.10)	n.d. 125	4.45 (0.21)	100	*	*
July 2019	5.39 (0.37) 6.02 (0.11)	<15 n.d.	*	*	6.74 (0.40)	500

*: not sampled; n.d.: not detected. Standard deviations are indicated in parenthesis.Four CRB strains, representative of distinct colony morphologies, were further characterized for taxonomic identification and presence of selected antibiotic resistance determinants *(aac (6′)-Ib, blaTEM, bla CTX-M, qnrS, sul1, and intI1)* (Table 6 and Table 7). The species *Vagococcus fluvialis*, *Pediococcus pentosaceus*, *Aeromonas rivipollensis*, and *Escherichia coli* were identified. *A. rivipollensis* harbored four out of six antibiotic resistance determinants screened.

**Table 6 foods-09-01641-t006:** Isolates of ciprofloxacin-resistant bacteria identified.

Production Zone	May	June	July
Site 1	*	*Vagococcus fluvialis*	***
Site 2	*	*Aeromonas rivipollensis* *Pediococcus pentosaceus*	***
Site 3	*		*Escherichia coli*

*: Not detected.

**Table 7 foods-09-01641-t007:** Ciprofloxacin enrichment cultures: identification and antibiotic resistance genes.

Species (phylum)	Class 1 Integron	β-Lactam	β-Lactam	Quinolone	Aminoglycoside	Sulfonamide
	*IntI1* 280 bp	*blaTEM* 1080 bp	*blaCTX-M* 540 bp	*qnrS* 463 bp	*aac*(6) 482 bp	*sul1* 789 bp
*Vagococcus fluvialis*	-	-	-	-	+	-
*Pediococcus pentosaceus*	-	-	-	-	+	-
*Aeromonas rivipollensis*	+	-	+	+	+	+
*Escherichia coli*	+	+	-	-	-	-

+: present; -absent.

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
