# Peer review of "Identification of Emerging Hazards in Mussels by the Galician Emerging Food Safety Risks Network (RISEGAL). A First Approach"

_foods, 2020, doi:10.3390/foods9111641_

Round 1

Reviewer 1 Report

The manuscript can be splitted in different sections.

The first section, more important and sound is focused on detection the emerging hazards potentially associated with bivalve shellfish produced in Galicia.

The section on prioritization of emerging hazards could be reduced.

Additionally, an on-line survey was carried out. The on-line survey should be removed from the manuscript since very few answers were obtained with no statistical significance. 

2.1-2.3 The authors should explain their choice of considering online, non-scientific and scientific surveys.

2.5.4 Please specify if the Authors followed a standard protocol, and in case add an appropriate reference

Table 1: please specify what 1 or 2 asterisks mean (sampling frequency?)

line 377-389: write taxonomic names in italics, and check further in the manuscript

line 398: please provide a brief explanation for these results

Author Response

QUESTIONS AND RESPONSES FOR REVIEWER 1

Q1: The on-line survey should be removed from the manuscript since very few answers were obtained with no statistical significance. 

A1: Thanks for your comment. Concerning your objection to a low numbers of respondents (50), we would like to say that respondents were all experts (food inspectors, staff from the academia and researchers, and professionals from the bivalve molluscs-based food industry) in the subject. Expert elicitations generally consists of a similar number of respondents (see Hoelzer et al., 2012 and Sanabria et al., 2019). In scenarios of high uncertainty, outputs thus obtained permit to identify emerging problems of concern for the main actors involved.

Q2: 2.1-2.3 The authors should explain their choice of considering online, non-scientific and scientific surveys.

A2: The following sentence was included in line 104.: “To gather data from the different stakeholders concerned with the molluscs bivalve chain, data collection was carried out by these three complementary approaches”

Q3­:2.5.4 Please specify if the Authors followed a standard protocol, and in case add an appropriate reference.

A3: To identify the colonies, we have followed the same procedure described in Rodríguez-López et al. [11], based in rRNA gene sequence and analysis.

Q4: Table 1: please specify what 1 or 2 asterisks mean (sampling frequency?).

A4: Sorry, the number of asterisks corresponds with the number of samples points analyzed from each Ría and site of production. I have already changed the title of Table 1 to clarify.

Q5: line 377-389: write taxonomic names in italics, and check further in the manuscript.

A5: Thanks, I have already cheched it.

Q6: line 398: please provide a brief explanation for these results.

A6: The explanation is that their condition of emerging implies uncertainly in detection.

Reviewer 2 Report

The manuscript describes the process for emerging risk identification in Galicia (Spain). As such, is not properly a research paper nor a review (it is closer to a survey type article), but I feel it falls within the aims and scope of the journal.

Comments

Reference 1 I think you should use EFSA as an author (or Anonymous)

l52 contributing to though: probably misspelled, please check

l61 refs 4 and 5 are both on viruses, perhaps you should a review with a wider scope

l75 are you sure that reference 7 is the right one? It is related to oysters in Western France, not Galicia

l95-96 carried out a scientific survey... please revise

Table 1 which is the meaning of * as opposed to **?

l115 check use of italics in species names

Table 3: use English (not Spanish) style for question marks

Figure 3: I think it can be improved, either by ordering the risks in decreasing order based on median or by using box plots or some other sort of display to show variablity (as interquartile range); I think the degree of uncertainty / lack of consensus among the expert is also important here

l379 check the usage of italics

l430-432 check the usage of italics; identification of a larger number of isolates  and characterizationo f their antibiotic resistance profile would have been desirable  

Author Response

REVIEWER 2

Q1: Reference 1. I think you should use EFSA as an author (or Anonymous).

A: Done.

Q2. l52 contributing to though: probably misspelled, please check.

A: Thanks, corrected in the text.

Q3. l61 refs 4 and 5 are both on viruses, perhaps you should a review with a wider scope.

A: References 4 and 5 have been already changed.

Q4. l75 are you sure that reference 7 is the right one? It is related to oysters in Western France, not Galicia.

A: Thanks for your question. We have substituted reference 7 by new references added, 4 and 5.

Q4. l95-96 carried out a scientific survey... please revise.

A: We have changed the sentence.

Q5. Table 1 which is the meaning of * as opposed to **?

A: the number of asterisks correspond with the number of samples collected. I have already changed the title of the table.

Q6: l115 check use of italics in species names.

A: done.

Q7: Table 3: use English (not Spanish) style for question marks.

ANSWER: Thanks for your comment. I have revised english style.

Q8: Figure 3: I think it can be improved, either by ordering the risks in decreasing order based on median or by using box plots or some other sort of display to show variablity (as interquartile range); I think the degree of uncertainty / lack of consensus among the expert is also important here.

A: Sorry, we have been compared our figure with box plot graphic and we reasserted that expressing obtained results in terms of frecuency of responses facilitates its interpretation by the readers and permit to narrow down the responses sent by the experts in two intervals.

Q9: l379 check the usage of italics

A: Done.

Q10: l430-432 check the usage of italics; identification of a larger number of isolates  and characterization of their antibiotic resistance profile would have been desirable.

A: We used as isolation criteria different colony morphology and accordingly with this criteria, only four isolates could be distinguished. 

Reviewer 3 Report

The paper entitled “A first approach to identify emerging hazards in mussels by the Galician emerging food safety risks network (RISEGAL) is a very interesting paper that addresses a topical issue of remarkable interest not only for the scientific community but also for the various players along the food chain. The work is experimentally well conducted and well written. Only minor corrections were suggested.

The authors throughout the text should check and revise the name of microorganisms which should be reported in italic (lines 115, 350, 357, 364, 365, 379,381, 393, 430, 431, 436, 438, 439).

The authors should report in the results and discussion section the results of the presence of antibiotic residues. If the results are negative, the authors should comment these results.

Author Response

REVIEWER 3

Q1: The authors throughout the text should check and revise the name of microorganisms which should be reported in italic (lines 115, 350, 357, 364, 365, 379,381, 393, 430, 431, 436, 438, 439).

ANSWER: Done

Q2:

ANSWER: I have included a sentence in section 3.4.2. to clarify this.

Reviewer 4 Report

This is an interesting article aimed to identify and prioritize the emerging hazards potentially associated with bivalve shellfish produced in Galicia. Identification of emerging risks is considered a priority for the European Food Safety Authority. The paper is well written and aims and methods are clearly indicated. Moreover, there are some needed minor revisions:

Line 61. Add more references about viral and bacterial diseases transmission related to mollusks bivalve consumption (e.g. “Ventrone, et al., 2013. Chronic or accidental exposure of oysters to norovirus: is there any difference in contamination? Journal of Food Protection”

Line 63. Add more references about the sanitary controls at production and retail levels (e.g. Pepe et al., 2012, Norovirus monitoring in bivalve molluscs harvested and commercialized in Southern Italy);

 Line 85. If possible, add a reference about Typeform.

Line 86. Please, specify which exactly arehealth professionals”

Line 110. About the sampling, please, specify:

- how many samples did you collected in total

- if you did a pool of several samples for each sampling location (e. g. and how many samples from the corners and from the center of the sampling area)

- if you considered the weather conditions before sampling and if they were different among sampling periods

- if you considered the seawater and environmental conditions (temperature, pH, ecc)

Line 294. Please, specify if the on-line survey was anonymous

Line 402.  Usually the viral contamination is higher in winter period. Please, specify if you choose the spring period (May, June, and July) for a specific reason.

Line 412.  About Italy, add the references “Fusco et al., 2019. Detection of hepatitis a virus and other enteric viruses in shellfish collected in the gulf of Naples, Italy” and “Suffredini et al., 2014. Quantitative and qualitative assessment of viral contamination in bivalve mollusc harvested in Italy. International Journal of Food Microbiology”)

Line 435. Since the European Regulation 2073/2005 assess that “the conventional faecal indicators are unreliable for demonstrating the presence or absence of NLVs and that the reliance on faecal bacterial indicator removal for determining shellfish purification times is unsafe practice”, did you find any relationship between E. coli and HEV shellfish contamination?

Author Response

REVIEWER 4

Q1: Line 61. Add more references about viral and bacterial diseases transmission related to mollusks bivalve consumption (e.g. “Ventrone, et al., 2013. Chronic or accidental exposure of oysters to norovirus: is there any difference in contamination? Journal of Food Protection”

Q2: Line 63. Add more references about the sanitary controls at production and retail levels (e.g. Pepe et al., 2012, Norovirus monitoring in bivalve molluscs harvested and commercialized in Southern Italy);

 Q3: Line 85. If possible, add a reference about Typeform.

Q4:Line 86. Please, specify which exactly are “health professionals”.

ANSWER: I have changed this by “public health inspectors”

Q5: Line 110. About the sampling, please, specify:

Q5.1. how many samples did you collected in total.

ANSWER: We have collected at total 10 samples,

Q5.2.If you did a pool of several samples for each sampling location (e. g. and how many samples from the corners and from the center of the sampling area).

ANSWER: Sorry, we collected mussels from different B production areas located in the galician coast. For location, please see Figure 1.

Q5.3.: if you considered the weather conditions before sampling and if they were different among sampling periods

ANSWER: No, we didn´t

- if you considered the seawater and environmental conditions (temperature, pH, ecc)

ANSWER. No, we didn´t

Q6: Line 294. Please, specify if the on-line survey was anonymous

ANSWER: I have added this information in the text. The condition of anonymous for the participants was optional in the programme.

Q7: Line 402.  Usually the viral contamination is higher in winter period. Please, specify if you choose the spring period (May, June, and July) for a specific reason.

ANSWER: No, there was not an specific reason for that choice.

Q8. Line 412.  About Italy, add the references “Fusco et al., 2019. Detection of hepatitis a virus and other enteric viruses in shellfish collected in the gulf of Naples, Italy” and “Suffredini et al., 2014. Quantitative and qualitative assessment of viral contamination in bivalve mollusc harvested in Italy. International Journal of Food Microbiology”).

ANSWER: Done, thanks.

Q9. Line 435. Since the European Regulation 2073/2005 assess that “the conventional faecal indicators are unreliable for demonstrating the presence or absence of NLVs and that the reliance on faecal bacterial indicator removal for determining shellfish purification times is unsafe practice”, did you find any relationship between E. coli and HEV shellfish contamination?

ANSWER: Sorry, we didn’t check that correlation in this study.

Reviewer 5 Report

A study was conducted to identify human health hazards associated with bivalves in Galicia. The research is very interesting and sound. This paper will be a welcome addition to the scientific literature. The only suggestions for improvement are editorial in nature.

  1. Line 52: delete “though”
  2. Line 70: change “of” to “from”
  3. Line 76: change “cost” to “coast”
  4. Line 146: change “the wide” to “their wide”
  5. Line 147: insert “they” before “represented”
  6. Line 148: change “it was” to “we”
  7. Line 183: why was the upper phase discarded and then re-extracted? Please clarify.
  8. Line 196: change “acid formic” to “formic acid”
  9. Line 202: change “is” to “was”
  10. Line 210: change “elutes is” to “eluted was”
  11. Line 248: 5 L of liquid was added to a 1.5 mL tube; obviously, this is not possible. Please clarify.
  12. Line 251: this statement is not clear.
  13. Line 303: should read: “…to Q2, which…”
  14. Line 371: should read: “…exposed to and be…”
  15. Line 379: genus and species should be italicized. Also, on lines 381 and 382 and lines 393 and 395 and lines 431 and 432 and lines 436, 438, and 439.
  16. Line 383: delete “…is attributed by some authors…”; redundant.
  17. Line 390: no new paragraph.
  18. Lines 429 and 430: genes should be italicized. Also, line 439.
  19. Line 450: insert “and” before “HEV”

Post-review comment (no response needed): It seems that epidemiological data is a good way to identify hazards and that collecting data on the prevalence, number, and type of hazard present in a serving of the food would be the cornerstone of a quantitative risk assessment that could integrate these data with other important factors (food handling practices, host resistance, food consumption behavior) that determine the actual public health risk of the consumer. In other words, the mere presence of a hazard does not indicate a public health risk; it depends on other important risk factors as well. This was eluted to in the present paper.

Author Response

REVIEWER 5

  1. Line 52: delete “though”. Amended.
  2. Line 70: change “of” to “from”. Amended.
  3. Line 76: change “cost” to “coast”. Amended
  4. Line 146: change “the wide” to “their wide”. Amended.
  5. Line 147: insert “they” before “represented”. Amended.
  6. Line 148: change “it was” to “we”. Amended.
  7. Line 183: why was the upper phase discarded and then re-extracted? Please clarify. It was included a new sentence in the parragraph to clarify.
  8. Line 196: change “acid formic” to “formic acid”. Amended
  9. Line 202: change “is” to “was”. Amended.
  10. Line 210: change “elutes is” to “eluted was”. Amended.

11.Line 248: 5 L of liquid was added to a 1.5 mL tube; obviously, this is not possible. Please clarify. Amended

  1. Line 251: this statement is not clear.

A:I am not sure to which statement are you referred to. Sorry, lines numbers cited by the reviewers does not correspond with line numbers of the new version of the text.

  1. Line 303: should read: “…to Q2, which…”. Amended.
  2. Line 371: should read: “…exposed to and be…”Amended.
  3. Line 379: genus and species should be italicized. Also, on lines 381 and 382 and lines 393 and 395 and lines 431 and 432 and lines 436, 438, and 439. Amended.
  4. Line 383: delete “…is attributed by some authors…”; redundant.Amended
  5. Line 390: no new paragraph. Amended.
  6. Lines 429 and 430: genes should be italicized. Also, line 439. Amended.
  7. Line 450: insert “and” before “HEV”. Amended.

Round 2

Reviewer 1 Report

Ok, the manuscript can be published as it is